# Polyacrylonitrile Composites Blended with Asphalt as a Low-Cost Material for Producing Synthetic Fibers: Rheology and Thermal Stability

**DOI:** 10.3390/ma17235725

**Published:** 2024-11-22

**Authors:** Artem V. Pripakhaylo, Alexei A. Tsypakin, Anton A. Klam, Andrei L. Andreichev, Andrei R. Timerbaev, Oksana V. Shapovalova, Rustam N. Magomedov

**Affiliations:** 1Semenov Research Center for Chemical Physics, Russian Academy of Sciences, Kosygin St. 4, 119991 Moscow, Russia; andrei.timerbaev@mail.ru (A.R.T.); shapovalova@chph.ras.ru (O.V.S.); 2UMATEX, Varshavskoe Sh. 46, 115230 Moscow, Russia; a.tsipakin@umatex.ru (A.A.T.);

**Keywords:** polyacrylonitrile, asphalt, rheological properties, plasticizing additive

## Abstract

The results of rheological studies and thermal analysis of polymer compositions based on polyacrylonitrile copolymers (PAN) of different molecular weights and asphalt isolated by n-pentane solvent deasphalting are presented. It was found that the asphalt content in mixtures with PAN at the level of 10–30 wt.% improves the rheological properties of the polymer composite melt. In particular, the temperatures of extrusion and molding of fibers tend to reduce, and the time during which the melt retains its rheological characteristics necessary for extrusion is notably increased, from 43 to 92 min. Thermal analysis by DSC revealed no effect of asphalt additive in an amount of up to 30 wt.% on radical PAN cyclization and the subsequent stage of fiber stabilization. Our study proved the possibility of preparing polymer composites based on PAN and asphalt suitable for extrusion and eventual molding of continuous filaments of synthetic fibers with reduced cost of production.

## 1. Introduction

The main type of raw materials from which the production of textile and high-strength fibers is carried out is polyacrylonitrile copolymers (PAN). The PAN fibers are generally produced via an expensive solution-molding technology, which is also environmentally unfriendly due to the use of large amounts of toxic organic solvents [1]. On the contrary, fiber formation from PAN melt needs no solvents and eliminates polymer losses upon increasing the production speed.

Nowadays, more than in years before, there is a growing concern of polymer chemists to reduce the production cost of synthetic fibers, including those based on PAN, which is widely used both in the textile industry and in the production of reinforcing materials. Perhaps the most straightforward approach is employing PAN blends with cheaper components. For instance, for the production of low-cost fibers, Ding et al. proposed to employ lignin in the amount of 10–45 wt.% of the total weight of PAN [2,3]. The resulting composite fibers are characterized by increased fire resistance, while carbon fibers (CFs) possess high tensile strength. However, neither PAN fibers blended with lignin nor the respective CFs can reach the mechanical properties of parental PAN fibers [4].

Another class of promising additives includes a cheap by-product of solvent deasphalting (SDA), asphalt, characterized by high softening temperatures [5,6,7]. The conversion of SDA products into the CF directly, without any energy-intensive preparation steps, has been the focus of research by Saad et al. [8]. and Leistenschneider and coworkers [9]. Asphalts of pentane (C5) SDA and C5-asphaltenes were used in these studies as inexpensive feedstocks for producing the CF. It is worth noting that n-pentane provides the final product as dry solid granules or powder [10]. In addition, the resulting C5-asphaltenes feature a high softening temperature (more than 170 °C). On the other hand, to boost the physical and mechanical properties of CF, it is required to treat a pitch precursor at the stage of stabilization with HNO_3_ followed by heat treatment [11]. The obtained CFs are distinct in an elastic modulus of 71 GPa and tensile strength of 1130 MPa, which, however, are inferior to similar parameters of CFs based on an individual PAN. Thus, asphalt is an additive to PAN that holds the promise of achieving satisfactory physical and mechanical indicators of CFs while reducing the total cost of their production.

However, the polymeric compositions with different additive contents may not be easily subject to subsequent fiber formation. One of the main methods for the preliminary assessment of the melt processability/spinnability is rheology research. The melt-spinning ability is determined by the thermal rheological behavior under theoretical forming conditions. The temperature-dependent complex viscosity profile, as well as the accumulation and loss modulus (G′ and G″, respectively), are useful factors to evaluate the minimum values of process parameters that a PAN composition can achieve before the cross-linking between the nitrile groups reaches critical values [12,13].

Our work is aimed at studying the rheological and thermal behaviors of PAN-based compositions blended with asphalt and assessing their prospective for molding the polymer fibers by the spin-melt method. The results described below give impetus to developing the theoretical basis of the molding of plasticized melts intended to produce cheap synthetic fibers based on PAN.

## 2. Materials and Methods

### 2.1. Materials

The PAN polymers of the same composition but different molecular weights, 55 kDa (PAN-1) and 91 kDa (PAN-2), were obtained via radical (precipitation) copolymerization. Differing significantly in molecular weight, the synthesized PAN copolymers feature different rheological properties. PAN-1 becomes viscous when heated to 172 °C and can therefore be potentially melt-formed into fiber. For PAN-2, the transition to a viscous state is possible only for a short period of time at 184 °C, which does not allow the formation of the respective fiber. As a plasticizer, the asphalt of C5-SDA of vacuum residue was used, with a C7-asphaltene content of 52 wt.%.

### 2.2. C5-SDA Procedure

Asphalt was produced in a high-pressure autoclave with a volume of 2 L, equipped with an anchor mixer and an external electric heating jacket. An electric belt heater heated a drain valve installed at the bottom of the vessel to maintain the minimum permissible viscosity of the asphalt during sampling. After the vacuum residue sample (oil refinery, Russia) was cooled to room temperature, heavy oil feedstock and n-pentane (99.0% wt, Ekos-1 JSC, Moscow, Russia) were added to the autoclave to avoid solvent evaporation. The autoclave was sealed, heated to 170 °C, and stirred. The agitator’s rotation speed during extraction was 600 rpm. If necessary, after reaching the operating extraction temperature, the pressure was slowly elevated to 50 bar by adding additional amounts of solvent using a plunger pump. The total ratio of n-pentane to the vacuum residue sample was 6/1 (*v*/*v*). The extraction time was counted from the moment the set values of temperature, pressure, and rotation speed of the agitator were established. The extraction process was completed after 30 min, and the mixture was kept without stirring for 30 min. Upon phase separation, the asphalt was removed through a bottom valve and brought to a constant mass in a drying oven at a temperature not exceeding 120 °C [14]. The content of asphaltenes insoluble in n-heptane was determined according to the IP 143 (ASTM D 6560) method [15].

### 2.3. Polymer Blend Preparation

Composite samples were prepared by mechanical grinding and mixing of solid PAN and asphalt to form a homogeneous dispersed product using a porcelain mortar and pestle. Prior to mixing, PAN and asphalt were dried in a drying oven at a temperature of 100 °C until a constant mass. Samples were molded under pressure of about 5 tons to produce a round tablet with a diameter of 1.5 cm and a thickness of 1.2–1.3 mm. Figure 1 shows a comparison of polymer composites obtained by grinding and pressing.

### 2.4. Polymer Characterization

The molecular weight of polymers was determined by gel-permeation chromatography according to ISO 16014-1:2019 [16] and ISO 16014-3:2019 [17] on a Shimadzu Prominence LC-20 instrument (Shimadzu, Kyoto, Japan) equipped with a RID-20A differential refractometer. The elution was performed with DMF containing 1 wt.% LiBr at 60 °C. DSC analyses were performed by ISO 11357-5:2013 [18] using a DSC 214 Polyma analyzer (Netchzt, Hanau, Germany) with heating from 50 to 400 °C at a rate of 10 °C/min. in the air atmosphere. A modular rheometer Anton Paar MSR 102 (Anton Paar GmbH, Graz, Austria), operating in the range from 140 to 190 °C, was used for rheological studies.

Temperature dependencies of the storage and loss modules were recorded in two consecutive modes. According to mode (1), the sample was heated at a rate of 2 °C/min in the range of 165 to 190 °C so that the temperature of the phase transition from the elastic to the viscoelastic state (T_1_) can be determined from the equality of G′ and G″ at the tangent of the mechanical loss angle (tg δ) = 1. Upon heating, the moduli become equal when the material transitions from an elastic to viscoelastic state. Therefore, T_1_ can be considered as the melting point of the material. As the material is further heated, tg δ increases and the next significant point is tg δ = 1.2. It is reached at a certain temperature value (T_1.2_ thereafter) when the material becomes suitable for melt spinning and obtaining continuous filaments from the melt. In mode (2), the sample was subjected to isothermal heating at T_1.2_ to estimate the melt lifetime, i.e., the time at which the material is characterized by tg δ ≥ 1.2.

## 3. Results and Discussion

### 3.1. Thermal Analysis

Of utmost interest in the DSC thermograms of polymers is the characteristic exothermic peak in the region of 220–300 °C [19,20], which portrays the complex reactions of thermal oxidation, as well as intra- and intermolecular cyclization (mainly by CN groups). The latter is the basis for the thermal stabilization of PAN fibers and their further high-temperature processing. As can be seen in Figure 2, the DSC peak of initial PAN has a characteristic temperature range (220–400 °C), with a maximum at about 320 °C, and asphalt additives from 10 to 30 wt.% exert no significant effect on its shape and position along the temperature scale.

The results of the DSC analysis are presented in Table 1. It should be noted that the decrease in the peak area (or reaction enthalpy) for PAN–asphalt mixtures is probably due to the endothermic effect of the cracking reaction of petroleum components [21,22]. The thermogram of the original asphalt is characterized by an endothermic effect at temperatures above 370 °C associated with the cracking of petroleum asphaltenes, which is confirmed in the literature [23,24].

### 3.2. Rheological Studies

For these studies, we employed the oscillation test mode with controlled shear deformation and constant angular velocity. The advantage of such testing is the ability of measurements without destroying the sample structure and the detailing of rheological characteristics by isolating elastic and viscous components. The characteristics in question were the storage modulus G′ (elastic component), the loss modulus G″ (viscous component), and their ratio, i.e., the tangent of the mechanical loss angle tg δ, determined at a strain value of 1%, a thickness of 1 mm, and a constant angular velocity of 31.5 rad/s.

For asphalt, the rheological properties were assessed in the range of 140–190 °C. The lower limit of the temperature range was selected experimentally via a gradual decrease in the initial temperature until the temperature of the intersection of the G′ and G″ curves was attained. It is important to note that at temperatures close to and above 190 °C, PAN-based samples are subject to a release of gaseous products due to oxidation and cyclization processes [25]. Such a phenomenon seriously distorts the rheological properties and makes the processes of extrusion and spinning technologically impossible (let alone a comparison between PAN and asphalt). Therefore, we avoided rheological measurements beyond the specified temperature range.

Rheological curves defining the temperature behavior of G′ and G″ are shown in Figure 3.

Throughout the entire temperature range, the G″ exceeds the G′, which is indicative of the viscous state of asphalt. At the same time, both G″ and G′ values decrease with increasing temperature, while the G″/G′ ratio gradually increases from 1.05 at 140 °C to 2.58 at 180 °C. These observations imply the absence of thermal cross-linking, destruction, and other adverse temperature effects on the rheological properties in the range under scrutiny [26]. At temperatures between 170 and 180 °C, the complex viscosity of asphalt reduces from 1288 to 569 Pa × s, remaining below the same parameter of the original PAN samples (see Table 2). This opens the opportunity of using asphalt as a plasticizing additive.

Figure 4 shows rheological curves for the PAN-1 and PAN-2 polymers prepared as described in the Materials and Methods section. The minimum temperature was chosen as 165 °C because at lower temperatures the samples are in a solid state, which prevents them from gaining the dimensions necessary for testing. The rheological characteristics of PAN-1 and PAN-2 are fairly different. Specifically, the intersection point for G′ and G″, corresponding to the transition from an elastic to a viscous state [12], is reached at 172 and 184 °C for PAN-1 and PAN-2, respectively. The critical value of tg δ, i.e., 1.2, is observed for PAN-1 at 176 °C (T_1.2_), at which the melt displays a “lifetime” of 43 min. In contrast, for PAN-2, this value is not achievable over the entire range studied, and the lifetime has not been determined. Our preliminary studies have shown that at tg δ ≤ 1.2, PAN melts cannot form continuous fibers, although the extrusion process remains an opportunity.

For polymer composites based on PAN-1, rheological curves are presented in Figure 5. The temperature of the point of intersection, as well as T_1.2_, tends to shift toward lower values with an increase in asphalt content, thereby confirming its plasticizing action. A similar effect of asphalt is also evident from the data on the lifetime of polymer compositions (Table 3).

For example, the addition of 20 wt.% of asphalt leads to a more than two-fold increase in the lifetime, up to 87 min. However, a further increase in asphalt content has no significant effect. Apparently, longer lifetimes of the melt are mainly associated with decreased T_1.2_ and deceleration of the cyclization processes by CN groups. This enables the melt to retain its rheological properties and fiber-formation potential for a longer time. It should be mentioned that the complex viscosity of melts with different asphalt contents does not notably differ from the viscosity of the original PAN-1 melt. Therefore, when lowering the melt viscosity is an issue, using asphalt as an additive makes an untoward choice.

To further assess the prospect of using asphalt for blending PAN-based melts, we evaluated the rheological parameters for higher molecular weight PAN-2. For the corresponding polymer compositions, similar rheological curves were recorded. Since PAN-2 has a softening point higher than PAN-1 (due to its higher molecular weight), the initial temperature was fixed at 170 °C, while the maximum temperature remained the same, 190 °C. The rheological data obtained are listed in Table 4.

Obviously, asphalt additives to PAN-2 favor the melt lifetime, holding the trend of gradually reducing T_1_ and T_1.2_. However, this effect seems insufficient for utilizing these composites for spin melting. Even for the melt fortified with 30 wt.% of asphalt, the lifetime is much shorter than 45 min, which is the minimum time required to maintain the extrusion and molding processes uninterrupted.

## 4. Conclusions

The principal possibility of using asphalt obtained by SDA of vacuum residue as an external plasticizer of PAN melts is demonstrated. In the selected temperature region, asphalt does not adversely affect the rheological properties of the polymer material and reduces the temperature of transition to a viscous state and eventually, the temperature of fiber formation of polymer composites. This extends the lifetime of the melt and should improve its moldability. Another anticipated advantage of asphalt compared to other plasticizers (ionic liquids, glycols, propylene carbonate, etc.) is that it does not need removal after the fiber is obtained. However, the main benefit of the proposed blending approach is perhaps due to the low cost of oil residues and the SDA process itself, which would inevitably reduce the total cost of production of PAN-based fibers. Our ongoing research is directed to the optimization of composition and rheological properties of PAN–asphalt composites, as well as to achieving insight into the extrusion, fibrillation, and drawing of fibers based on the optimum blends and the assessment of their applicability in various industries (automotive, civil engineering, production of sporting goods and various equipment, etc.).

## Figures and Tables

**Figure 1 materials-17-05725-f001:**
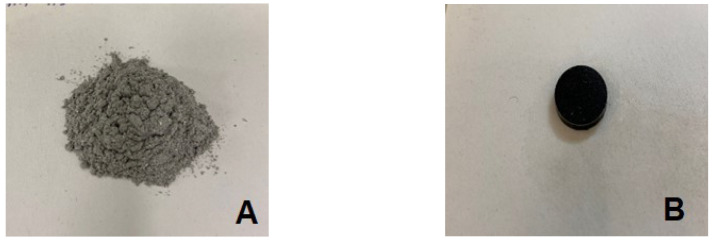
Images of the polymer composite after (**A**) grinding and (**B**) pressing.

**Figure 2 materials-17-05725-f002:**
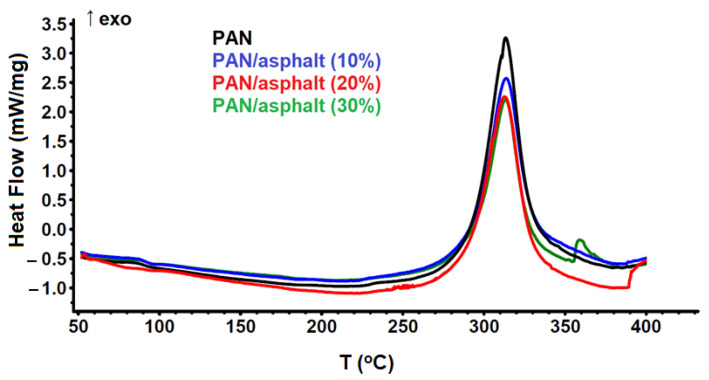
DSC-thermograms of the initial PAN and polymer composites with asphalt.

**Figure 3 materials-17-05725-f003:**
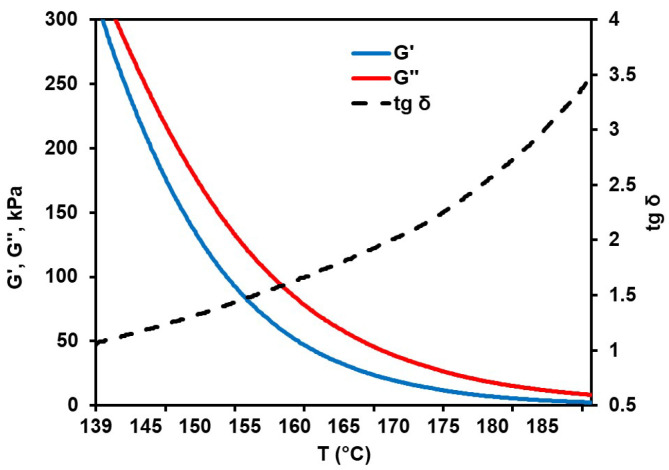
Rheological curves of asphalt.

**Figure 4 materials-17-05725-f004:**
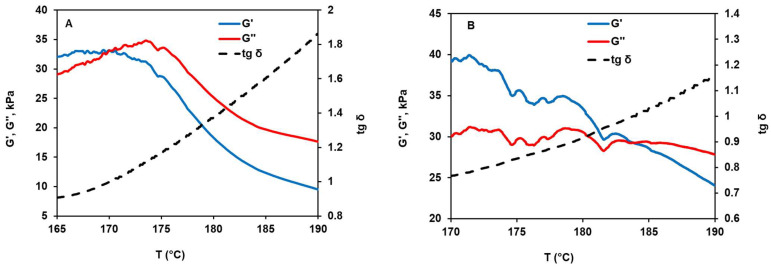
Rheological curves of PAN-1 (**A**) and PAN-2 (**B**).

**Figure 5 materials-17-05725-f005:**
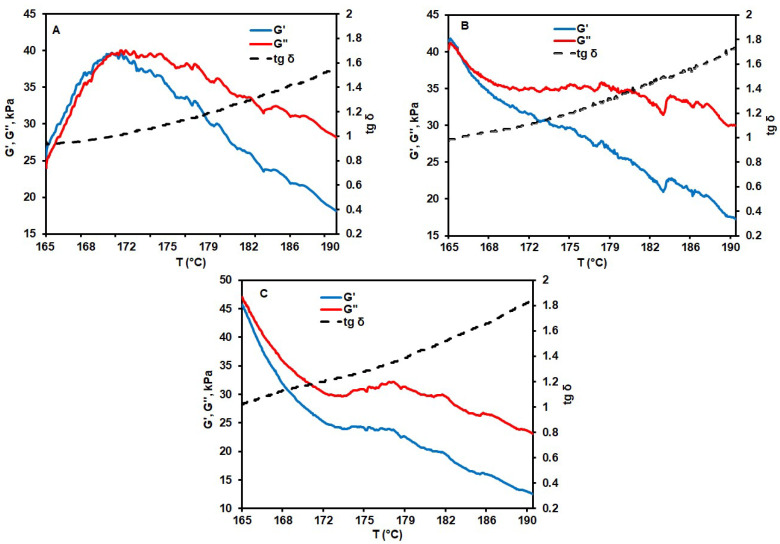
Rheological curves of PAN-1 composites with (**A**)—10; (**B**)—20; and (**C**)—30 wt.% asphalt.

**Table 1 materials-17-05725-t001:** Results of DSC analysis of initial PAN-1 and polymer composites with asphalt.

Sample	Peak Onset Temperature [°C]	Peak Temperature [°C]	Peak Area [J/g]
PAN-1	227.9	331.4	623.8
PAN-1 + 10% asphalt	227.5	312.5	547.4
PAN-1 + 20% asphalt	225.8	313.6	523.7
PAN-1 + 30% asphalt	224.2	313.3	476.6

**Table 2 materials-17-05725-t002:** Viscosity data.

Sample	Complex Viscosity [Pa × s]	
	170 °C	180 °C
Asphalt	1288	569
PAN-1	1487	970
PAN-2	2620	1980

**Table 3 materials-17-05725-t003:** Rheological parameters of PAN-1 composites.

Sample	T_1_ [°C] ^a^	T_1.2_ [°C]	Lifetime at T_1.2_ [min]	Complex Viscosity at T_1.2_ [Pa × s]
PAN-1	172	176	43	1363
PAN-1 + 10% asphalt	170	175	56	1229
PAN-1 + 20% asphalt	167	173	87	1431
PAN-1 + 30% asphalt	164	172	92	1390

^a^ At tg δ = 1.

**Table 4 materials-17-05725-t004:** Rheological parameters of PAN-2 composites.

Sample	T_1_ [°C] ^a^	T_1.2_ [°C]	Lifetime at T_1.2_ [min]	Complex Viscosity at T_1.2_ [Pa × s]
PAN-2	184	-	-	-
PAN-2 + 10% asphalt	181	188	3.5	1520
PAN-2 + 20% asphalt	171	180	9.0	2128
PAN-2 + 30% asphalt	169	178	15.3	2316

^a^ At tg δ = 1.

## Data Availability

The original contributions presented in the study are included in the article, further inquiries can be directed to the corresponding author.

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
