# Peer review of "Polyacrylonitrile Composites Blended with Asphalt as a Low-Cost Material for Producing Synthetic Fibers: Rheology and Thermal Stability"

_materials, 2024, doi:10.3390/ma17235725_

Round 1
Reviewer 1 Report
Comments and Suggestions for Authors
2.1. Materials
It is not clear and detailed enough. Are Materials bought? Or obtained in the laboratory by sinthesis? What producers? Give more details. Is it composite or not? How it is prepared?
2.2. SDA Procedure
What is the SDA Procedure? Authors are familiar with this but readers may not. Write the whole name not just the acronym for the first time. And give more details about this procedure.
2.3 Polymer Blend Preparation
Be more detailed, with what equipment how? This is so modest explanation. One sentence is too modest a description of sample preparation.
Line: 80-82
Sentence Could not start with a number. Not convenient. Redefine sentence. Also, this sentence seems not to belong here in methods but in materials.
Figure S1.
Figure S1. should have a y-axis Physical quantity noted. Name or notation missing.
Heat low per unit of mass.
Line: 86-87
This condition is not clear, clarify it or expand with more sentences. What are G` and G`` and tg d, the first time appear in the manuscript? Be more detailed and reader-friendly.
Line: 87-88
Does this sentence have a connection with the previous one? It looks like it should have but it is not clear. More explanation missing.
Line: 88 – 89
With one's studies? Add references.
Line: 98 – 100
When is peak observed and defined it is odd to give a range. More continent is to give maximum value or peak value. This range is so broad. What is possition of the peak? In the next sentence is the term position.
Figure 1.
Add physical quantity name or label - y-axis.
Line: 129 – 131
Be more precise, which module values? Clarify it.
Line: 133 – 135
Brackets missing or Pa x s.
Line: 139 – 140
Modest description in Materials and methods.
Line 145 - 146
How lifetime is assessed? Clarify it.
Line 153 – 154
Term values is more convenient than magnitude.
Line 175 – 176.
T1 is not defined in the text, what is it? First time appearing in the manuscript.
As the manuscript already is short and lacks details I recommend merging the supplementary to the main manuscript at appropriate places and discarding the suplementry.
Author Response
|
Comments 1: 2.1. Materials It is not clear and detailed enough. Are Materials bought? Or obtained in the laboratory by sinthesis? What producers? Give more details. Is it composite or not? How it is prepared? |
|
Response 1: We added the requested information to the revised text (changes marked in red).
|
|
Comments 2: 2.2. SDA Procedure What is the SDA Procedure? Authors are familiar with this but readers may not. Write the whole name not just the acronym for the first time. And give more details about this procedure. |
|
Response 2: Corrected. The acronym is introduced on P. 1.
Comments 3: 2.3. Polymer Blend Preparation Be more detailed, with what equipment how? This is so modest explanation. One sentence is too modest a description of sample preparation. Response 3: Improved.
Comments 4: Figure S1. should have a y-axis Physical quantity noted. Name or notation missing. Heat low per unit of mass. Response 4: This figure is removed from the final version.
Comments 5: Line: 86-87 This condition is not clear, clarify it or expand with more sentences. What are G` and G`` and tg d, the first time appear in the manuscript? Be more detailed and reader-friendly. Response 5: Done. Parameters G` and G`` are defined on P. 2.
Comments 6: Line: 87-88 Does this sentence have a connection with the previous one? It looks like it should have but it is not clear. More explanation missing. Response 6: Corrected.
|
|
Comments 7: Line: 88 – 89 With one's studies? Add references. Response 7: These studies remain a commercial secret (patented by the UMATEX).
|
|
Comments 8: Line: 98 – 100 When is peak observed and defined it is odd to give a range. More continent is to give maximum value or peak value. This range is so broad. What is possition of the peak? In the next sentence is the term position. Response 8: Corrected.
|
|
Comments 9: Figure 1. Add physical quantity name or label - y-axis. Response 9: Added (note that the figure number is 2 in the revised manuscript).
|
|
Comments 10: Line: 129 – 131 Be more precise, which module values? Clarify it. Response 10: Done.
|
|
Comments 11: Line: 133 – 135 Brackets missing or Pa x s. Response 11: Used only in the tables.
Comments 12: Line: 139 – 140 Modest description in Materials and methods. Response 12: Extended.
Comments 13: Line 145 - 146 How lifetime is assessed? Clarify it. Response 13: We added the requested clarification (the time at which the material is characterized by tg δ ≥ 1.2).
Comments 14: Line 153 – 154 Term values is more convenient than magnitude. Response 14: Changed.
Comments 15: Line 175 – 176. T1 is not defined in the text, what is it? First time appearing in the manuscript.. Response 15: Defined (on P. 3).
Comments 16: As the manuscript already is short and lacks details I recommend merging the supplementary to the main manuscript at appropriate places and discarding the suplementry. Response 16: Done
|

Reviewer 2 Report
Comments and Suggestions for Authors
The research study focuses on investigation of usage asphalt as external plasticizer of polyacrylonitrile-based fibers and characterization of thermal, viscoelastic and rheological properties of the composite. The paper could be interesting for the Materials readership. However, the article is suffering from insufficient literature review, lack of comparison the obtained results with literature.
There are some concerns that need to be addressed:
1) Please indicate the purity (in %), source (Supplier name) for all compounds and solvents used in this study and if they were purified/dried prior to use.
2) There was error in opening of Supporting Information archive, so these data was not possible to check.
3) What was the gas atmosphere used for the DSC study?
4) The authors did not provide comprehensive literature review on the investigated topic, therefore there is lack of comparison the obtained results:
e.g.
10.3390/app8101818
10.1016/j.carbon.2024.119300
10.1016/B978-0-443-13623-8.00009-5
Author Response
|
Comments 1: Please indicate the purity (in %), source (Supplier name) for all compounds and solvents used in this study and if they were purified/dried prior to use. |
|
Response 1: We added the qualification of chemicals used in the revised manuscript.
|
|
Comments 2: There was error in opening of Supporting Information archive, so these data was not possible to check. Response 2: We moved the whole Supporting Information section to the main text as advised reviewer 1.
Comments 3: What was the gas atmosphere used for the DSC study?. Response 3: Just the air atmosphere (added to the text).
Comments 4: The authors did not provide comprehensive literature review on the investigated topic, therefore there is lack of comparison the obtained results 10.3390/app8101818, 10.1016/j.carbon.2024.119300, 10.1016/B978-0-443-13623-8.00009-5. Response 4: We extended the introduction section by adding some references, including the work by Karaaslan et al. [10.1016/j.carbon.2024.119300]. However, this is the only of the mentioned references that is releated to our study
|

Reviewer 3 Report
Comments and Suggestions for Authors
This work investigates the rheological and thermal behaviors of PAN-based compositions blended with asphalt, aiming to develop a cheaper method for producing synthetic fibers based on PAN. The results show that asphalt can be used as a plasticizing additive, reducing the complex viscosity and enabling the formation of melts suitable for fiber production. The distinct rheological characteristics of PAN-1 and PAN-2 polymers are also compared, highlighting their different transition temperatures and "lifetime" under specific conditions.
This research work involves polymer compositions. The topic falls within the scope of Journal Material. However, there are several points that should be noted as follows:
1. The manuscript is very short. Recommend that the authors extend several sections and include more detailed descriptions of their research process.
2. The manuscript contains some typos, such as in the title of Section 1, "1. . Introduction". Suggest the authors carefully proofread the manuscript before resubmission.
3. Provide more background in Section 1. The authors could offer additional context on the current state of PAN-based fiber production and the challenges in existing methods.
4. Clarify and expand the experimental methods in Section 2. The authors should elaborate on the experimental procedures, especially the SDA procedure and polymer characterization steps, including specific parameters and conditions used.
5. Improve the context for the figures. By providing more detailed descriptions—including specific values and trends—the figures will be more informative.
6. Include a discussion section. The authors should provide a detailed comparative analysis with previous literature, highlighting specific differences and their implications.
7. Enhance the conclusion. Suggest discussing specific areas for future research, such as the long-term stability of the asphalt-plasticized PAN blends and potential applications in fiber manufacturing.
The English could be improved to more clearly express the research.
Author Response
|
Comments 1: 1. The manuscript is very short. Recommend that the authors extend several sections and include more detailed descriptions of their research process. |
|
Response 1: We followed this advice by detailing the experimental section (as also recommended reviewer 1) and by merging the main text with the supplementary material.
|
|
Comments 2: 2. The manuscript contains some typos, such as in the title of Section 1, "1. . Introduction". Suggest the authors carefully proofread the manuscript before resubmission. |
|
Response 2: Done.
Comments 3: 3. Provide more background in Section 1. The authors could offer additional context on the current state of PAN-based fiber production and the challenges in existing methods. Response 3: We revised the Introduction and added two full paragraphs (in red face) to respond this comment.
Comments 4: 4. Clarify and expand the experimental methods in Section 2. The authors should elaborate on the experimental procedures, especially the SDA procedure and polymer characterization steps, including specific parameters and conditions used. Response 4: Done (see the changes marked in red).
Comments 5: 5. Improve the context for the figures. By providing more detailed descriptions—including specific values and trends—the figures will be more informative. Response 5: Given the similarity of figures (cf. Figures 2-4), we figure that more details are not really necessary.
Comments 6: 6. Include a discussion section. The authors should provide a detailed comparative analysis with previous literature, highlighting specific differences and their implications. Response 6: We believe that after text modifications, including the addition of supplementary material, there is no need for more extended discussion of results.
|
|
Comments 7: 7. Enhance the conclusion. Suggest discussing specific areas for future research, such as the long-term stability of the asphalt-plasticized PAN blends and potential applications in fiber manufacturing. Response 7: We added a few industrial applications where the asphalt-plasticized PAN blends hold great promise. |

Round 2
Reviewer 1 Report
Comments and Suggestions for Authors
I want to thank the authors for their detailed and precise response and clarification of the manuscript. I wish you all the best in your further research and life.
Reviewer 2 Report
Comments and Suggestions for Authors
The authors made a great job in the manuscript reduction, especially by clarification of the experimental part. I satisfied with the changes made and the responses given.
Reviewer 3 Report
Comments and Suggestions for Authors
The authors have addressed the points in accordance with the comments.
Comments on the Quality of English LanguageThe English could be improved to more clearly express the research.